# Role of Sclerostin in Cardiovascular System

**DOI:** 10.3390/ijms26104552

**Published:** 2025-05-09

**Authors:** Ning Zhang, Luyao Wang, Xiaofei Li, Xin Yang, Xiaohui Tao, Hewen Jiang, Yuanyuan Yu, Jin Liu, Sifan Yu, Yuan Ma, Baoting Zhang, Ge Zhang

**Affiliations:** 1School of Chinese Medicine, Faculty of Medicine, The Chinese University of Hong Kong, Hong Kong 999077, China; ningzhang001@cuhk.edu.hk (N.Z.); hewenjiang@cuhk.edu.hk (H.J.); yusifan@126.com (S.Y.); 2Guangdong-Hong Kong-Macao Greater Bay Area International Research Platform for Aptamer-Based Translational Medicine and Drug Discovery (HKAP), Hong Kong 999077, China; lxf410516@163.com (X.L.); yangxin9613@gmail.com (X.Y.); taoxiaohui122@163.com (X.T.); yuyuanyuan@hkbu.edu.hk (Y.Y.); liujin@hkbu.edu.hk (J.L.); mayuan@hkbu.edu.hk (Y.M.); zhangge@hkbu.edu.hk (G.Z.); 3Law Sau Fai Institute for Advancing Translational Medicine in Bone & Joint Diseases (TMBJ), Hong Kong Baptist University, Hong Kong 999077, China

**Keywords:** sclerostin, cardiovascular system, clinical data, nonclinical data, molecular understanding

## Abstract

Sclerostin, encoded by the *SOST* gene, is a novel bone anabolic target for bone diseases. Humanized anti-sclerostin antibody, romosozumab, was approved for treatment of postmenopausal osteoporosis by the US Food and Drug Administration (FDA), but with a black-box warning on cardiovascular risk. The clinical data regarding cardiovascular events from various pre-marketing and post-marketing studies of romosozumab were inconsistent. Overall, the cardiovascular risk of sclerostin inhibition could not be excluded. The restriction of romosozumab in patients with cardiovascular disease history would be necessary. Moreover, genome-wide association study (GWAS) analyses of *SOST* variants revealed inconsistent results of the association between *SOST* variations and cardiovascular diseases. Further research incorporating larger sample sizes and functional analyses are necessary. In analyses of serum/tissue sclerostin levels in patients with cardiovascular diseases, the results were controversial but indicated an association between sclerostin and the presence/severity/outcomes of cardiovascular diseases. Nonclinical studies in rodents indicated the inhibitory effect of sclerostin on inflammation, aortic aneurysm, atherosclerosis, and vascular calcification. Sclerostin loop3 participated in the inhibitory effect of sclerostin on bone formation, while the cardiovascular protective effect of sclerostin was independent of sclerostin loop3. Macrophagic sclerostin loop2–apolipoprotein E receptor 2 (ApoER2) interaction participated in the inhibitory effect of sclerostin on inflammation in vitro. Sclerostin in human aortic smooth muscle cells participated in the reduction in calcium deposition. The role of sclerostin in cardiovascular system deserves further investigation.

## 1. Introduction

Sclerostin acts as an internal antagonist of the bone anabolic Wnt signaling pathway, and it is a novel bone anabolic target for the treatment of postmenopausal osteoporosis. A monoclonal antibody against sclerostin, romosozumab, has been demonstrated to substantially enhance bone mineral density (BMD) and decrease the risk of nonvertebral and vertebral fractures [1,2,3,4]. However, serious cardiovascular adverse events were reported in the romosozumab group, both in the phase III active-controlled fracture study in postmenopausal women with osteoporosis at high risk (ARCH) and another placebo-controlled study evaluating the efficacy and safety of romosozumab in treating men with osteoporosis (BRIDGE) [3,4]. Thus, romosozumab has been approved to reverse established osteoporosis in postmenopausal women by the US FDA and the European Medicines Agency (EMA), but with a black-box warning on its label declaring that individuals who had experienced a stroke or heart attack within the last 12 months should not be administered with romosozumab due to the potential risk of heart attack, stroke, and cardiovascular mortality. In a post-marketing cardiovascular safety evaluation of romosozumab, the incidence of major cardiovascular events (MACEs) in Japanese patients was significantly elevated after treatment with romosozumab [5,6]. However, the cardiovascular safety data from TriNetX, mostly from the USA and Western Europe, showed that the incidence of MACEs was significantly lower in the romosozumab group than that in the parathyroid hormone (PTH) analogs group [7]. The outcome of cardiovascular events after treatment with romosozumab were controversial in the clinic. Several nonclinical studies investigated the role of sclerostin in the cardiovascular system [8,9,10,11]. However, it still deserves to be further studied. In this review, we summarized pre-marketing and post-marketing cardiovascular safety evaluations of romosozumab, discussed the association between *SOST* variants and cardiovascular diseases, and integrated the data analyzing the association between sclerostin and the presence/severity/outcomes of cardiovascular diseases. Furthermore, we exhibited the preclinical studies of the role of sclerostin in the cardiovascular system and also gave insights into the molecular understanding of the role of sclerostin in the cardiovascular system.

## 2. Methods

Literature Search Strategy

The search and analysis of the scientific literature were carried out using the databases PubMed, Scopus, Web of Science, and Google Scholar. The following keywords and their combinations were used in the search: “sclerostin inhibition,” “randomized controlled trial,” “cardiovascular safety”, and “sclerostin mechanisms”.

Inclusion Criteria

Scientific literature published in English up to March 2025 was selected based on the following criteria: (a) studies of clinical trials in which subjects were treated with romosozumab, (b) meta-analysis involving multiple large-scale clinical trials of romosozumab treatment, (c) post-marketing surveillance evaluating cardiovascular safety of romosozumab, (d) genetic studies investigating the association between sclerostin variations and cardiovascular diseases, (e) clinical investigations examining the association between sclerostin and the presence/outcome/severity of cardiovascular diseases, and (f) preclinical research elucidating the role of sclerostin in the cardiovascular system.

Exclusion Criteria

Clinical trials on treatment with romosozumab without cardiovascular safety evaluations and preclinical studies focusing on the role of sclerostin out of the cardiovascular system were excluded.

## 3. Pre-Marketing and Post-Marketing Cardiovascular Safety Evaluation of Therapeutic Sclerostin Antibody

### 3.1. Pre-Marketing Cardiovascular Safety Evaluation

The preliminary safety information for the humanized anti-sclerostin monoclonal antibody, romosozumab, stemmed mainly from the FRAME (the fracture study in postmenopausal women with osteoporosis) [2], ARCH [3], and BRIDGE [4] clinical trials. In both the ARCH and the BRIDGE clinical trials, the incidence of adjudicated serious cardiovascular events significantly increased in the romosozumab group [2,3,4].

In the first year (double-blind period) of the ARCH trial, an imbalance with 12 additional adjudicated serious cardiovascular events was observed in the romosozumab group (50 adjudicated serious cardiovascular events, 2.5%) compared to the alendronate (ALN) group (38 adjudicated serious cardiovascular events, 1.9%) (odds ratio [OR], 1.31; 95% confidence interval [CI], 0.85–2.00) (Table 1) [3]. Cardiac ischemia events were recorded by 16 patients (0.8%) in the romosozumab group and 6 (0.3%) in the ALN group (OR, 2.65; 95% CI, 1.03–6.77), while cerebrovascular events were reported by 16 patients (0.8%) in the romosozumab group and 7 (0.3%) in the ALN group (OR, 2.27; 95% CI, 0.93–5.22) [3]. During the open-label ALN treatment period, the imbalance with 12 additional serious cardiovascular events in the romosozumab group throughout the first year was maintained but did not increase further [3]. Numerically, the incidence of cardiovascular death (romosozumab 2.8% vs. ALN 2.7%) and cardiac ischemia events (romosozumab 1.5% vs. ALN 1.0%) was higher among the women treated with romosozumab during the entire two-year study period, whereas the incidence of cardiac failure (romosozumab 0.6% vs. ALN 1.1%) was lower, when compared to the women treated with ALN. Khosla et al. concluded that the imbalance of the adjudicated serious cardiovascular events found in the ARCH study between the romosozumab group and the ALN group could be the possible cardiovascular protective effect of ALN [12,13]. However, in meta-analysis of the cardiovascular events in the ARCH study and other randomized controlled trials, there was no evidence for the cardiovascular protective action of ALN [14,15].

In the BRIDGE study, a higher incidence of cardiovascular events (cardiac ischemia and cerebrovascular events) was reported in the romosozumab group than the placebo group (Table 1) [4]. Cardiac ischemic events occurred in 3 of 163 (1.8%) men in the romosozumab group, and no case was reported in the placebo group [4]. Cerebrovascular events occurred in 3 of 163 (1.8%) men in the romosozumab group and in 1 of 81 men (1.2%) in the placebo group [4]. Adjudicated cardiovascular serious adverse events occurred in 8 of 163 (4.9%) men in the romosozumab group and 2 of 81 (2.5%) men in the placebo group [4].

In the FRAME trial, in the first 12 months, adjudicated serious cardiovascular events occurred in 41 of 3576 women (1.1%) in the placebo group and in 44 of 3581 (1.2%) women in the romosozumab group (Table 1) [2]. In the second 12 months, the drug in both groups changed to denosumab. During the whole two years, adjudicated serious cardiovascular events occurred in 79 of 3576 women (2.2%) in the placebo group and in 82 of 3581 (2.3%) women in romosozumab group [2]. This seemed to be inconsistent with the results of ARCH and BRIDGE. However, the data for the cardiac ischemic events and cerebrovascular events in the FRAME trial were not shown. Details of adjudicated serious cardiovascular events in the FRAME trail deserve to be further studied.

Another randomized, placebo-controlled clinical trial for evaluating the effect of romosozumab on hip fractures (NCT01081678) [16] also determined cardiovascular events. Acute myocardial infarction (MI) occurred in 3 of 238 patients (1.3%) in the romosozumab group and 1 of 87 patients (1.1%) in the placebo group. Cerebrovascular events occurred in 2 of 238 patients (0.8%) in the romosozumab group and 0 of 87 patients in the placebo group [16]. In a 6-month clinical trial conducted in Korea (NCT02791516) (n = 67), cardiac disorder occurred in 1 of 34 (2.9%) patients in the romosozumab group [17]. The administration time was much shorter than the 12-month treatment period recommended by the US-FDA. Moreover, in a 12-month clinical trial conducted in Japan, romosozumab (n = 26) and denosumab (n = 25) were compared for the treatment of severe osteoporosis in rheumatoid arthritis patients; no adjudicated serious cardiovascular events were observed in the romosozumab group [18]. The limited sample size might constrain the statistical power.

The meta-analysis of cardiovascular events in 4298 individuals from the ARCH and BRIDGE studies indicated that romosozumab significantly increased the occurrence rate of cardiac ischemic events (OR, 2.98; 95% CI, 1.18 to 7.55; *p* = 0.02) and cerebrovascular events (OR, 2.15; 95% CI, 0.94 to 4.92; *p* = 0.07), when compared to the ALN or placebo group [19]. The addition of the FRAME data in the meta-analysis indicated that romosozumab also significantly increased the occurrence rate of cardiac ischemic events (OR, 1.54; 95% CI, 0.90 to 2.64; *p* = 0.12), cerebrovascular events (OR, 1.44; 95% CI, 0.80 to 2.58; *p* = 0.22), and MACEs (OR, 1.39; 95% CI, 0.98 to1.98; *p* = 0.07), when compared to the control group [19]. Inconsistently, another meta-analysis conducted by Jonathan et al. [20] analyzed the data from the clinical trials, including the FRAME [2], ARCH [3], BRIDGE [4], NCT01081678 [16], STRUCTURE [21], NCT00896532 [22], and NCT01992159 [23] trials. The results of the meta-analysis suggested that romosozumab did not significantly increase the risk of MACE (OR, 1.14; 95% CI, 0.83 to 1.57; *p* = 0.54) or cardiovascular death (OR, 0.92; 95% CI, 0.53 to1.59; *p* = 0.71) [20]. Notably, the serious cardiovascular adverse events were not recorded in the STRUCTURE trial. The bone anabolic potential of romosozumab and teriparatide were compared for the treatment of osteoporosis [21]. In addition, both NCT00896532 and NCT01992159 were phase II clinical trials in which bone anabolic efficacy was the endpoint assessed. Data on the side effects on the cardiovascular system were not included in the STRUCTURE, NCT00896532, and NCT01992159 clinical trials. Meta-analyses need to incorporate the clinical trials in which adverse events on the cardiovascular system were reported to improve the reliability and accuracy of the analysis.

### 3.2. Post-Marketing Cardiovascular Safety Evaluation

The humanized anti-sclerostin monoclonal antibody, romosozumab, was first and commercially launched on the Japanese market for the therapy of osteoporosis in patients at high risk of fracture in March 2019 (Amgen Astellas BioPharma Release). In a press statement about the drug’s side effects released in July 2019, Astellas-Amgen revealed that within the first three months of the drug’s commercial availability, there were 11 occurrences of severe cardiovascular adverse events, including three deaths [5]. After six months of marketing, 68 serious cardiovascular events, including 16 deaths, were reported in October 2019. The possibility that these occurrences were associated with the romosozumab could not be excluded (Astellas Pharma Report). After the post-marketing report on romosozumab revealed unforeseen severe cardiovascular adverse events, a red-box warning was incorporated in the prescription instruction of romosozumab, stating “Judge depending on the benefits and risks. If a patient experiences cardiovascular events during therapy, ask patients to consult related doctors.” [5].

Annika et al. extracted and analyzed all cases reported between January 2019 and December 2020 from the US Food and Drug Administration Adverse Event Reporting System (US FAERS) [6]. In comparison to all other drug cases in the FAERS database, higher rates of MACEs were determined in subjects treated with romosozumab. This was consistent with the results in the BRIDGE and ARCH clinical trials as well as the above post-marketing result from Japan [3,4,5]. Regional variations were noted in the analysis of data from the US FAERS. In this pharmacovigilance analysis reporting 206 MACE outcomes, 164 (79.6%) originated from Japan and 41 (19.9%) were derived from the US. Compared to the USA, the population from Japan had a greater tendency to be male and older and had more serious outcomes, such as hospitalization or intervention [6] (Table 2). The author indicated that patients with prior cardiovascular risk factors might be at higher risk of MACEs, especially the aged (>80 years old) and those with co-reported cardiovascular medications [6].

In addition, another study in Japan collected 702,072 reports from the Japanese Adverse Drug Event Report (JADER) database (a database similar to the US FAERS) and analyzed the association between cardiovascular events and osteoporotic drugs, including romosozumab, bisphosphonates, teriparatide, and denosumab [24]. Higher incidence of cardiac ischemia events and higher incidence of cerebrovascular events were only detected in patients treated with romosozumab [24]. The author further investigated the risk factors for cardiac ischemia events and cerebrovascular events in the patients receiving romosozumab (n = 859). The data showed that the incidence of cardiac ischemia events significantly increased in romosozumab users with a history of hypertension (OR, 1.6; 95% CI 1.0 to 2.7; *p* = 0.047) and cardiac disease (OR, 5.9; 95% CI 3.5 to 9.9; *p* < 0.01). Moreover, the incidence of cerebrovascular events was significantly higher in the presence of hypertension (OR, 2.6; 95% CI 1.7 to 3.9; *p* < 0.01) and cerebrovascular disease (OR, 2.7; 95% CI 1.2 to 6.2; *p* = 0.02) [24]. Together, hypertension might be a significant risk factor for the cardiac ischemia events and cerebrovascular events in the patients receiving romosozumab [24].

Joshua et al. [7] analyzed the cardiovascular data from TriNetX using the propensity-score matching (PSM) algorithm and compared the cardiovascular safety of romosozumab against parathyroid hormone (PTH) analogs. The patients who had been diagnosed with an acute cardiovascular event during the prior year were excluded from both cohorts. The incidence of MACEs was significantly lower in the romosozumab group (158 patients, 2.8%) than that in the PTH group (211 patients, 3.8%, *p* = 0.003). In detail, myocardial ischemic events were recorded by 31 patients (0.6%) in the romosozumab group and 58 (1.0%) in the PTH group (*p* = 0.003), cerebrovascular events were reported by 56 patients (1.0%) in the romosozumab group and 79 (1.4%) in the PTH group (*p* = 0.037), and cardiovascular deaths were reported by 83 patients (1.5%) in the romosozumab group and 104 (1.9%) in the PTH group (*p* = 0.099) [7].

In the above studies recorded by the US FEARS database [6], the JADER database [24], and another clinical study in Japan [5], the patients in Japan with a history of cardiovascular disease were not restricted for the use of romosozumab. The data from the above studies consistently suggested that the incidence of MACEs in Japanese patients was significantly elevated after treatment with romosozumab [5,6,24]. In the study recorded by the TriNetX database, the patients in Western Europe and the USA who had been diagnosed with an acute cardiovascular event during the prior year were excluded from the use of romosozumab [7]. The incidence of MACEs was significantly lower in the romosozumab group when compared to that in the PTH analog group [7]. This implied that the protective effect of sclerostin on the cardiovascular system was conditional. Sclerostin could participate in protecting the cardiovascular system for individuals with a history of cardiovascular disease. Accordingly, the restriction of romosozumab in patients with cardiovascular disease history would be necessary.

## 4. The Association Between *SOST* Variants and Cardiovascular Diseases

In the UK Biobank, the meta-analysis by Bovijn et al. of the cardiovascular events of two *SOST* variants (rs7209826 and rs188810925) associated with increased bone mass and reduced fracture risk showed that genetically lowered sclerostin led to a higher risk of cardiovascular events [19]. However, Holm et al. critiqued Bovijn et al.’s study and stated that the significance threshold (*p* = 0.02) used in the genetic analysis was not stringent enough [25]. In response, Bovijn et al. cautioned against the sole reliance on strict *p*-value thresholds for dismissing potential biological effects, especially when considering the broader context of consistent findings across multiple datasets. They emphasized that their findings, although not reaching conventional GWAS thresholds (*p* = 5 × 10^−8^), were consistent with the evidence from randomized controlled trials, suggesting a genuine cardiovascular risk associated with sclerostin inhibition [26]. Moreover, Holm et al. critiqued this and stated that the associations observed between *SOST* variants and cardiovascular risk might not be directly attributable to the *SOST* variants themselves [25]. They pointed out the possibility of other variants in the adjacent region having stronger associations with cardiovascular traits (triglyceride and systolic blood pressure), leading to confounding effects. Bovijn et al. argued that while risk factors of blood lipids or blood pressure were considered, the *SOST* variants could potentially be associated with cardiovascular risk through their effect on BMD, rather than confounding effects by local linkage disequilibrium [26]. However, in the analysis by Holdsworth et al., three other *SOST* variants (rs9899889, rs1107748, and rs66838809) were associated with increased BMD and reduced fracture risk but exhibited no correlation with cardiovascular risk of MI or stroke [27], suggesting that the association between *SOST* variants and cardiovascular risk needs further study.

Further highlighting the assessment of cardiovascular risks, GWAS meta-analysis by Zheng et al. predicted that decreased sclerostin levels might elevate the risk of hypertension, myocardial infarction, and coronary artery calcification [28]. Staley et al. critiqued Zheng et al.’s study for using both cis and trans genetic variants in their mendelian randomization analysis, which could introduce confounding effects [29]. They noted that the corrected results did not meet the stringent multiple testing threshold and questioned the validity of the findings. Furthermore, Staley et al. found no significant association between lower sclerostin levels and cardiovascular risks in their additional analyses, suggesting that the original claims were overstated. They also emphasized the need for colocalization analyses to avoid false-positive results. In response, Zheng et al. leveraged newly available proteomic data from the SomaLogic and O-link platforms, which include sclerostin measurements, now accessible from DeCode and UK Biobank [30]. By analyzing these data across more than 100,482 individuals, they identified two significant single-nucleotide polymorphisms (SNPs) (rs66838809 and rs1107748) in the *SOST* region, with robust evidence linking lower sclerostin levels to increased coronary artery disease risk. The findings withstood the Bonferroni correction threshold (*p* = 0.0052), reinforcing the association between *SOST* variants and cardiovascular risk. We have genetically validated the association between low levels of sclerostin and improved bone health and reduced fracture risk. A recently published study proved a causal relationship between reduced sclerostin (*SOST* variants: rs7220711 and rs66838809) and an increased risk of coronary artery disease (OR = 1.85 [1.12, 3.06]) and type 2 diabetes (OR = 1.35 [0.98, 1.87] by Mendelian randomization (MR) analysis [31]. Also, the results highlighted a potential causal relationship between genetically predicted lower levels of sclerostin and myocardial infarction (OR = 1.35 [0.98, 1.87]) and hypertension (OR = 1.03 [0.99, 1.07]) [31]. While only people of European descent were included in this research, the findings of this study may not be applied to other groups with diverse genetic backgrounds.

The above studies explored the relationship between *SOST* variations and cardiovascular diseases. However, the findings in these studies are heterogeneous, as not all *SOST* variants were significantly linked to cardiovascular risk, indicating the necessity to clarify these associations. This complexity in genetic associations underscores the need for further research, including large-scale randomized controlled trials, to establish the definite association between cardiovascular risk and *SOST* variations.

According to Wang et al.’s comprehensive GWAS analysis in the UK Biobank, three variants (rs879666342, rs886052981, and rs765435662) in the loop2 region of the *SOST* gene were associated with cardiac dysrhythmias, precordial pain, and peripheral vascular disease, respectively [32]. Although the analysis provided valuable insights, the association between these variants and cardiovascular risk did not meet the conventional GWAS significance threshold (*p* = 5 × 10^−8^). It was evident that the limited sample size constrained its statistical power. Thus, the limited sample size highlighted the challenge of stringent correction methods for multiple testing, which may filter out potentially significant variants. Future research should aim to integrate additional datasets to expand the sample size of cardiovascular disease patients with *SOST* variations, thereby enhancing the reliability and accuracy of the analysis. Nonetheless, the potential association between *SOST* variations in the coding region and cardiovascular risk indicated that these variations might influence cardiovascular diseases through changes in protein conformation rather than solely affecting gene expression levels. This underscores the importance of coding region variants in understanding direct impacts on protein function and disease mechanisms. Therefore, conducting further functional studies is necessary to elucidate how such variants influence cardiovascular risk through the changes in protein’s structural conformation and interaction.

## 5. The Association Between Sclerostin and Cardiovascular Diseases

### 5.1. The Association Between Sclerostin and the Presence of Cardiovascular Diseases

#### 5.1.1. Serum Sclerostin Levels in Patients with Vascular and Valve Calcification

The association between serum sclerostin and vascular calcification appeared controversial. Claes et al. found that non-dialysis chronic kidney disease (ND-CKD) patients with aortic calcification had higher serum sclerostin levels than those without aortic calcification in univariate analysis (*p* = 0.0009) [33]. However, in multivariate logistic regression analysis, lower serum sclerostin levels were independently associated with aortic calcification (*p* = 0.04) [33]. Furthermore, Koos et al. found that patients with aortic valve calcification had higher serum sclerostin levels than those without aortic valve calcification (*p* < 0.001) [34]. Similarly, some studies showed that CKD patients with vascular calcification [35,36,37,38,39] and heart valve calcification (*p* < 0.05) [37,40] had higher serum sclerostin levels compared to noncalcified counterparts, respectively. Higher serum sclerostin levels were also detected in males undergoing coronary artery bypass graft (CABG) with aberrant coronary artery calcification (CAC) scores (CAC > 100) (*p* = 0.005) [41], when compared to those with normal CAC scores.

#### 5.1.2. Serum Sclerostin Levels in Patients with Arterial Stiffness

In comparison with patients with low arterial stiffness, the serum sclerostin levels were determined to be higher in multiple clinical cohorts with high arterial stiffness, including renal transplantation recipients (brachial-ankle pulse wave velocity [baPWV] >14.0 m/s) (*p* = 0.001) [42], postmenopausal women (carotid-femoral PWV [cfPWV] > 9 m/s) (*p* = 0.03) [43], hypertensive patients (cfPWV > 10 m/s) (*p* < 0.001) [44], end-stage renal disease patients (cfPWV > 10 m/s) (*p* = 0.0001) [45], and type 2 diabetes mellitus (T2DM) patients (cfPWV > 10 m/s) (*p* < 0.001) [46]. In contrast to the above clinical observations, Gaudio et al. reported that individuals with high arterial stiffness had lower serum sclerostin levels than those with low arterial stiffness (*p* < 0.05) [47] in adult healthy outpatient subjects.

#### 5.1.3. Serum Sclerostin Levels in Patients with Atherosclerosis

Serum sclerostin levels were significantly higher in hemodialysis patients with carotid artery atherosclerosis (*p* = 0.016) [48] and T2DM patients with atherosclerosis (*p* = 0.006) [49], compared to their counterparts without atherosclerosis. After adjustment for major confounding variables, higher serum sclerostin levels were found to be independently associated with atherosclerosis in T2DM patients (*p* = 0.012) [49]. Furthermore, T2DM patients with abnormal carotid artery media thickness (CIMT, C-IMT > 0.9 mm) also exhibited higher serum sclerostin levels in comparison to those with normal CIMT and healthy subjects (*p* < 0.001) [50]. Additionally, serum sclerostin levels were found to be independently associated with CIMT in T2DM (*p* = 0.017) [50] and prevalent hemodialysis patients (*p* = 0.03) [48], respectively.

Nevertheless, in a study of African American men and women with T2DM, Register et al. did not observe an association between serum sclerostin levels and calcified atherosclerotic plaques in women, except for an inverse correlation with carotid calcified atherosclerotic plaques in men (*p* = 0.03) [51]. Additionally, in postmenopausal women with T2DM, Gaudio et al. observed an inverse association between serum sclerostin levels and CIMT (*p* = 0.006) [52].

#### 5.1.4. Serum Sclerostin Levels in Patients with Peripheral Arterial Disease

In elderly people (>65 years old) [53] and hypertensive patients [54], individuals with peripheral arterial disease (ABI < 0.9) had higher serum sclerostin levels than those without peripheral arterial disease, respectively (*p* < 0.001). Moreover, multivariate logistic regression analysis revealed a significant independent correlation between serum sclerostin levels and peripheral arterial disease in both elderly (*p* = 0.008) [53] and hypertensive subjects (*p* = 0.002) [54].

#### 5.1.5. Serum Sclerostin Levels in Patients with Acute Ischemic Stroke

A study by He et al. showed that serum sclerostin levels were significantly higher, both in patients with large-artery atherosclerotic stroke (*p* < 0.001) and those with small-artery occlusion stroke (*p* < 0.001) compared with controls, and no significant difference was discovered between the above two stroke subtypes [55].

The above studies indicated the association between sclerostin and the presence of cardiovascular diseases (vascular valve calcification, arterial stiffness, atherosclerosis, peripheral arterial disease, and acute ischemic stroke) (Table 3).

**Table 3 ijms-26-04552-t003:** The association between serum sclerostin and the presence of cardiovascular diseases.

Patient Population	Cardiovascular-Related Diseases		Association (Serum Sclerostin Levels and Cardiovascular Diseases)	*p*-Value	References
ND-CKD patients (N = 154)	vascular calcification	aortic calcification	Positive (Univariate analysis)	0.0003	Claes et al. [33]
			Negative (Multivariate analysis)	0.04	
ND-CKD patients (N = 241)		coronary artery calcification	Positive	<0.001	Morena et al. [36]
CKD patients (N = 162)		coronary artery calcification	Positive	0.001	Ma et al. [38]
CKD patients (N = 97)		vascular calcification	Positive	<0.05	Lv et al. [39]
Males undergoing CABG (N = 61)		coronary artery calcification	Positive	0.005	Kim et al. [41]
CKD patients (N = 80)		abdominal aortic calcification	Positive	<0.05	Elarbagy et al. [37]
	valve calcification	aortic valve calcification	Positive	<0.05	
Patients with aortic valve calcification (N = 115)		aortic valve calcification	Positive	<0.001	Koos et al. [34]
CKD patients (N = 110)		heart valve calcification	Positive	<0.05	Ji et al. [40]
Renal transplant recipients (N = 82)	arterial stiffness		Positive	0.001	Hsu et al. [42]
Postmenopausal women (N = 149)			Positive	0.03	Hampson et al. [43]
Hypertensive patients (N = 105)			Positive	<0.001	Chang et al. [44]
End-stage renal disease patients (N = 194)			Positive	0.0001	Wu et al. [45]
T2DM patients (N = 125)			Positive	<0.001	Yang et al. [46]
Adult healthy outpatient subjects (N = 67)			Negative	<0.05	Gaudio et al. [47]
Prevalent hemodialysis patients (N = 122)	atherosclerosis		Positive	0.016	Nephrology et al. [48]
T2DM patients (N = 78)			Positive	0.006	Morales-Santana et al. [49]
African American men/women with T2DM (N = 450)			No Association (in women)	-	Register et al. [51]
			Negative (in men)	0.03	
T2DM patients and controls (N = 70)	subclinical atherosclerosis		Positive	<0.001	Shalash et al. [50]
Postmenopausal women (N = 40)			Negative	0.006	Gaudio et al. [52]
Elderly persons (>65 years) (N = 68)	peripheral arterial disease		Positive	<0.001	Teng et al. [53]
Hypertensive patients (N = 92)			Positive	<0.001	Wang et al. [54]
Patients with acute ischemic stroke (N = 122)	large-artery atherosclerotic stroke		Positive	<0.001	He et al. [55]
	small-artery occlusion stroke				

Abbreviations: ND-CKD, non-dialysis chronic kidney disease; CKD, chronic kidney disease; T2DM, type 2 diabetes mellitus; CABG, coronary artery bypass graft; Note: “Positive” indicates a direct association (higher serum sclerostin levels associated with the presence of cardiovascular diseases), “Negative” indicates an inverse association, and “No Association” indicates no significant correlation found.

### 5.2. The Association Between Sclerostin and Severity/Outcomes of Cardiovascular Diseases

In CKD patients, higher serum sclerostin levels were associated with a lower severity of vascular calcification (*p* < 0.001) [35]. The association was consistent in renal transplant recipients (*p* = 0.0003) [56] and hemodialysis patients (*p* = 0.035) [57], respectively. Furthermore, increased serum sclerostin levels were associated with decreased cardiovascular events in CKD patients (*p* = 0.02) [58] and low short-term (18-month) cardiovascular mortality in dialysis patients [59]. In elderly patients with stable coronary artery disease undergoing percutaneous coronary intervention, patients with high serum sclerostin levels had a lower incidence rate of main adverse cardiovascular and cerebrovascular events (MACCEs) (*p* = 0.014) and better survival (*p* < 0.05) than those with low serum sclerostin levels [60].

Some studies showed a positive association between serum sclerostin levels and the severity of vascular calcification [34,43,61,62,63] and coronary artery disease [64], respectively. Higher serum sclerostin levels were also associated with an increased risk of cardiovascular mortality in dialysis patients [65,66,67,68] and in individuals with and without T2DM [69]. Additionally, Kanbay et al. observed a significant association between high serum sclerostin levels and both fatal and nonfatal cardiovascular events in ND-CKD patients, even after multiple statistical adjustments (*p* < 0.001) [70].

The above studies indicated the association between sclerostin and the severity/outcomes of cardiovascular diseases. The investigation of tissue levels of sclerostin in cardiovascular diseases may facilitate understanding of the role of sclerotin in the cardiovascular system.

### 5.3. The Tissue Levels of Sclerostin in Cardiovascular Diseases

A study by Krishna et al. showed that sclerostin levels within the human aortic aneurysm samples were significantly lower than those within the control samples from normal human abdominal aortas (*p* = 0.008) [8]. Miteva et al. found VSMC-specific sclerostin expression levels were significantly lower in the atherosclerotic plaques of patients with ischemic stroke than those of asymptomatic people (*p* < 0.0001) [71]. The above studies indicated that sclerostin in aorta or atherosclerotic plaques might exert an inhibitory effect on the progression of aortic aneurysm and atherosclerosis. Interestingly, Koos et al. observed that within the same cohort of patients with aortic valve calcification, sclerostin expression levels were significantly higher in calcified aortic valves compared to noncalcified control valves (*p* = 0.002) [34].

The above studies were controversial but indicated the association between sclerostin and the presence/severity/outcomes of cardiovascular diseases.

## 6. Molecular Understanding of the Role of Sclerostin in the Cardiovascular System

### 6.1. The Role of Sclerostin in Inflammatory Responses, Aortic Aneurysm, and Atherosclerosis of ApoE^−/−^ Mice

Consistent with the clinical results of significantly lower sclerostin levels within the human aortic aneurysm samples than those within normal human abdominal aortas [8], the sclerostin protein levels within the suprarenal aorta samples from *ApoE*^−/−^ mice that developed aortic aneurysm (AA) were significantly lower than those from mice that did not develop AA after angiotensin II (AngII) infusion [8]. Furthermore, it has been reported that the transgenic introduction of sclerostin in *ApoE*^−/−^ (*SOST^Tg^.ApoE*^−/−^) mice and the administration of recombinant sclerostin inhibited AngII-induced AA and atherosclerosis development. Compared to *ApoE*^−/−^ mice, *SOST^Tg^.ApoE*^−/−^ mice showed decreased serum inflammatory cytokines and chemokines and reduced inflammatory cell infiltration in the aortas [8]. Also, Yu et al. demonstrated that human full-length sclerostin knock-in (*hSOST^ki^*) suppressed the progression of AA and atherosclerosis and significantly decreased the expression of inflammatory cytokines and chemokines in *hSOST^ki^.ApoE^−/−^* mice [9]. Collectively, it indicated the protective role of sclerostin in the cardiovascular system of *ApoE^−/−^* mice.

In one nonclinical cardiovascular safety assessment of the rat sclerostin antibody (Scl-Ab) r13C7 (https://www.accessdata.fda.gov/drugsatfda_docs/nda/2019/761062Orig1s000MultidisciplineR.pdf, accessed on 6 June 2024), r13C7 increased the incidence of plaques with necrosis of all types (2–5) in ovariectomized (OVX) *ApoE*^−/−^ mice with a high-fat diet. Plaques with necrosis occurred in 32 of 39 aorta segments in the r13C7 group and in 24 of 39 aorta segments in the control group, suggesting a potential concern about the increased plaque instability during sclerostin antibody treatment. Additionally, r13C7 upregulated the local expression of inflammatory cytokines and chemokines such as interleukin-6 (IL-6), monocyte chemoattractant protein-1 (MCP-1), and intercellular adhesion molecule 1 (ICAM1) in the aortas of OVX *ApoE*^−/−^ mice with a high-fat diet, while it did not affect systemic markers of inflammation or endothelial activation. In the published report of this research by Turk et al., the local expression levels of IL-6, MCP-1, and ICAM1 at the aortas were not reported [72]. Inconsistently, Yu et al. found that Scl-Ab elevated systemic levels of inflammatory cytokines and chemokines, including IL-6, TNF-α, and MCP-1, and aggravated the progression of AA and atherosclerosis in *ApoE*^−/−^ mice with AngII infusion [9]. Notably, in contrast to the weekly therapeutic dose (25 mg/kg-50 mg/kg per week) used in the treatment of bone diseases in rodents, Turk et al. [72] used a lower weekly administration dose of Scl-Ab (10 mg/kg per week). Although at a low administration dose (10 mg/kg per week), Scl-Ab increased the incidence of plaques with necrosis of all types (2–5) in *ApoE*^−/−^-OVX mice with a high-fat diet. In the study by Yu et al. [9], Scl-Ac (25 mg/kg, twice weekly) significantly promoted the progression of AA and atherosclerosis and significantly increased the serum levels of inflammatory cytokines and chemokines in *ApoE*^−/−^ mice with AngII infusion. Taken together, the aggravative effect of sclerostin inhibition on inflammation and atherosclerosis could not be excluded. The role of sclerostin in inflammation, AA, and atherosclerosis deserves further study.

The central regions of sclerostin consist of three loop-like domains (loop1, loop2, and loop3) (Figure 1) [73]. Yu et al. and Wang et al. found that sclerostin loop2 might participate in the inhibitory effect of sclerostin on inflammation, AA, and atherosclerosis in *ApoE*^−/−^ mice (Figure 2a), while the above inhibitory effect of sclerostin was independent of sclerostin loop3 [9,10]. Furthermore, sclerostin loop2 was reported to bind to ApoER2 in macrophages. The binding of sclerostin loop2 to ApoER2 participated in the inhibitory effects of sclerostin on inflammatory responses in macrophages in vitro [32]. ApoER2, encoded by low-density lipoprotein receptor-related protein 8 (*LRP8*), is a transmembrane receptor of the low-density lipoprotein receptor (LDLR) family [74]. Clinical data demonstrated that the ApoER2-R952Q variant had a 2-fold greater susceptibility to cardiovascular diseases and was associated with the early onset of myocardial infarction (MI) [75]. Moreover, it was reported that ApoER2 could mediate the effect of ApoE on the conversion of the macrophages into anti-inflammatory M2 phenotypes, the induction of p38^MAP^ kinase activation, and the inhibition of inflammatory responses in vivo [76,77]. Importantly, the absence of ApoER2 may cause vascular inflammation and lead to the progression of lesions in atherosclerosis [78]. Whether sclerostin loop2–ApoER2 interaction could participate in the inhibitory effects of sclerostin on the inflammatory responses and atherosclerosis progression in vivo warrants further investigation.

### 6.2. The Role of Sclerostin in Vascular Calcification of Mice

During vascular calcification, vascular smooth muscle cells (VSMCs), which line the arterial wall, have been shown to undergo a phenotypic conversion into cells that resemble osteoblasts [79]. The Wnt signaling pathway has been validated to be involved in the progression of VSMC transdifferentiation and mineralization [79,80,81]. Thus, the Wnt pathway inhibitor, sclerostin, was hypothesized to influence the development of vascular calcification. De Maré et al. demonstrated that the absence of sclerostin significantly aggravated the development of vascular calcification in cardiac vessels following adenine-induced renal failure in *Sost*^−/−^ mice [82]. On the other hand, calcifications formed in the aorta and renal arteries in DBA/2J mice (a mouse model to develop ectopic calcifications) on a warfarin diet. Inhibition of sclerostin via sclerostin antibodies significantly accelerated the progress of vascular calcification in DBA/2J mice [82]. These data indicated that sclerostin could protect against vascular calcification in mice (Figure 2b). However, how sclerostin works in vascular calcification remains unclear. Calcium deposition occurred in the artery during the progression of vascular calcification [83]. A recent study discovered for the first time that overexpression of sclerostin in human primary aortic smooth muscle cells (HAoSMCs) significantly reduced calcium deposition (Figure 2c), suggesting the potential inhibitory role of sclerostin in vascular calcification [11].

The above in vivo and in vitro studies indicated that the protective effect of sclerostin on the cardiovascular system could not be excluded. Nevertheless, whether inhibition of endogenous levels of sclerostin could influence the inflammation and the progression of AA and atherosclerosis remains unknown. Thus, the effect of sclerostin deficiency on the cardiovascular system could be evaluated in *SOST*^−/−^*.ApoE*^−/−^ mice, and a more comprehensive understanding of the role of sclerostin in cardiovascular system could be revealed.

### 6.3. The Role of Sclerostin-Related Pathway in the Cardiovascular System

Sclerostin acts as an antagonist of the Wnt signaling pathway through its interaction with low-density lipoprotein receptor-related protein 5/6 (*LRP5/6*) [84]. In humans, it has been found that *LRP5* polymorphisms are associated with obesity [85], hypertension, and hypercholesterolaemia [86,87]. A family with characteristics of osteoporosis and the metabolic syndrome was shown to have a history of autosomal dominant early coronary artery disease (CAD). These characteristics were genetically linked to a missense mutation in *LRP6* [88]. In addition, it has been found that the expression of Wnt3a significantly increased in human coronary atherosclerotic plaques compared with healthy tissues [89]. Also, Wnt3a was involved in the pathogenesis of CAD by altering the VSMC phenotype and migration, implying a pathophysiological role of Wnt3a in the regulation of CAD [89].

### 6.4. The Role of Sclerostin and ApoE in Cardiovascular System

In the above-reported in vitro study in Section 6.1, the suppressive effects of sclerostin on inflammatory responses in macrophages were found to be dependent on sclerostin loop2–ApoER2 interaction [32]. It was reported that ApoE, another ligand binding to ApoER2, has anti-inflammatory effects by converting macrophages from the M1 (pro-inflammatory) phenotype to the M2 (anti-inflammatory) phenotype [76]. The *APOE* genotypes, including three alleles (e2, e3, and e4), exert a distinct impact on lipids and inflammation and could be associated with atherosclerosis and hypertension [90]. A very low ApoE circulating concentration in humans was related to early-onset atherosclerosis [91]. Also, the deletion of the ApoE gene in mice increased their susceptibility to atherosclerosis [92]. These data indicated that ApoE played an anti-inflammatory function by binding to ApoER2. Whether sclerostin and ApoE contribute to the cardiovascular protective effect via ApoER2 through a mutually compensating mechanism deserves further study to provide evidence for the prediction of the cardiovascular risk populations with sclerostin inhibition treatment. Moreover, taking up the above discussion on the GWAS analyses of the association between *SOST* variants and cardiovascular diseases, future GWAS studies might delve into the combined impact of *SOST* variation and *ApoE* variation on cardiovascular phenotypes.

## 7. Discussion and Conclusions

The results of the pre-marketing and the post-marketing safety evaluations of the therapeutic sclerostin antibody, romosozumab, were not consistent. In large-scale clinical trials, the incidence of cardiac ischemic events and cerebrovascular events was significantly increased after romosozumab treatment in ARCH and BRIDGE. In the FRAME trial, the adjudicated serious cardiovascular events were balanced between the placebo group and romosozumab group, while the detailed data of cardiac ischemic events or cerebrovascular events were not reported and deserve further investigation. In small-scale clinical trials (NCT01081678, n = 325; NCT02791516, n = 67), acute MI, cerebrovascular events, or cardiac disorder were found in the romosozumab group, but had no significant difference from those in the control group. In another small-scale trial in Japan (n = 51), no serious cardiovascular events were observed in the romosozumab group. The limited sample size might constrain the statistical power. One meta-analysis of cardiovascular events in patients from ARCH, BRIDGE, and FRAME showed that romosozumab significantly increased the incidence of cardiac ischemic events, cerebrovascular events, and MACEs. Another meta-analysis from ARCH, BRIDGE, FRAME, NCT01081678, STRUCTURE, NCT00896532, and NCT01992159 showed that romosozumab did not significantly increase the risk of MACEs or cardiovascular death. Notably, data on the side effects on the cardiovascular system were not included in the clinical trials of STRUCTURE, NCT00896532, and NCT01992159. Meta-analyses need to incorporate the clinical trials in which adverse events on the cardiovascular system were reported to improve the reliability and accuracy of the analysis. Furthermore, in the post-marketing clinical application of romosozumab in Japan (patients with cardiovascular disease history were not restricted for use), the incidence of MACEs was significantly elevated. In the post-marketing clinical application in Western Europe and the USA (patients with cardiovascular disease history were restricted), the incidence of MACEs was significantly lower in the romosozumab group than the PTH analog group. Accordingly, restriction of sclerostin inhibition in patients with cardiovascular diseases history would be necessary.

Genetically, some *SOST* variants (rs7209826, rs188810925, rs66838809, and rs1107748) (in the noncoding region) were reported to be associated with higher risk of cardiovascular diseases. Inconsistently, three other *SOST* variants (rs9899889, rs1107748, and rs66838809) (in the noncoding region) exhibited no correlation with myocardial infarction (MI) or stroke. In addition, three other variations (rs879666342, rs886052981 and rs765435662) in the coding region of the *SOST* gene were found to be associated with cardiac dysrhythmias, precordial pain, and peripheral vascular disease, respectively. Taking the above data, variations in the *SOST* gene might influence cardiovascular diseases through changes in protein conformation rather than solely affecting gene expression levels. Moreover, the debate surrounding the association between distinct *SOST* variants and cardiovascular diseases suggested further research incorporating larger sample sizes and functional analyses.

Clinically, in the studies evaluating the serum/tissue sclerostin levels in patients with and without cardiovascular diseases, the results were controversial but indicated the association between sclerostin and the presence/severity/outcomes of cardiovascular diseases. Furthermore, nonclinical studies investigating the effect of sclerostin on the cardiovascular system indicated the suppressive effect of sclerostin on the progression of atherosclerosis, AA, inflammation, and vascular calcification in mice.

Combining the results of the clinical studies and nonclinical studies, the role of sclerostin in the cardiovascular system, especially for individuals with cardiovascular history, could not be ignored and deserves further study.

## Figures and Tables

**Figure 1 ijms-26-04552-f001:**
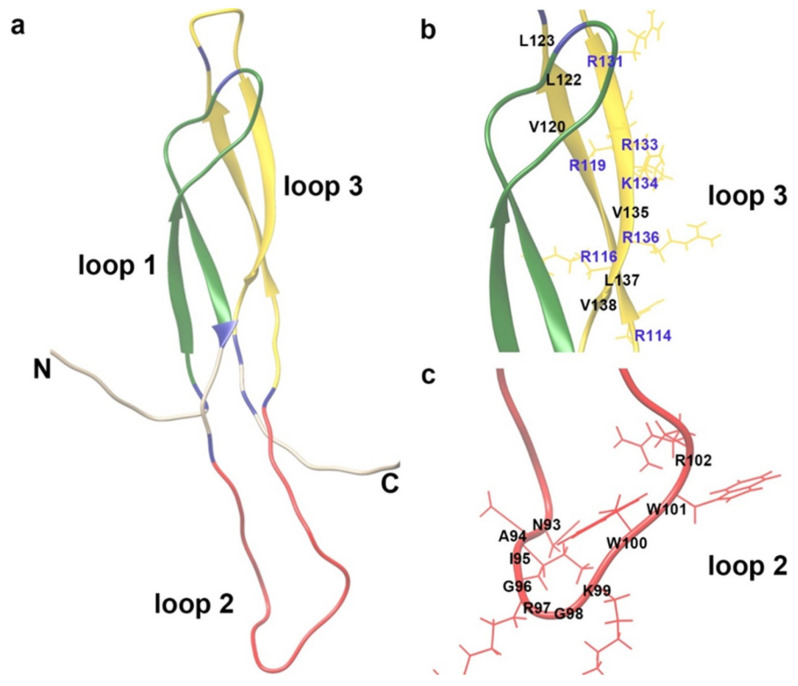
The illustration of three-dimensional structure of sclerostin. (**a**) Ribbon drawing of the solution structure of sclerostin (PBD ID: 2K8P). Long N- and C-terminal regions are highly flexible. Other residues formed three loop-like domains (green: loop 1; red: loop 2; yellow: loop 3). Six cysteine residues are highlighted in blue. (**b**) Potential active sites on loop 3 of sclerostin. Positive charged residues (K and R) and labeled in blue and their side chains are shown in sticks. Hydrophobic residues (V and L) are labeled in black. These residues are included in the positive target of virtual screening. (**c**) Motifs on loop 2 that bind to discovered sclerostin inhibitors. Residues N93-R102 are labeled in black, including potential interaction sites between sclerostin and inhibitors.

**Figure 2 ijms-26-04552-f002:**
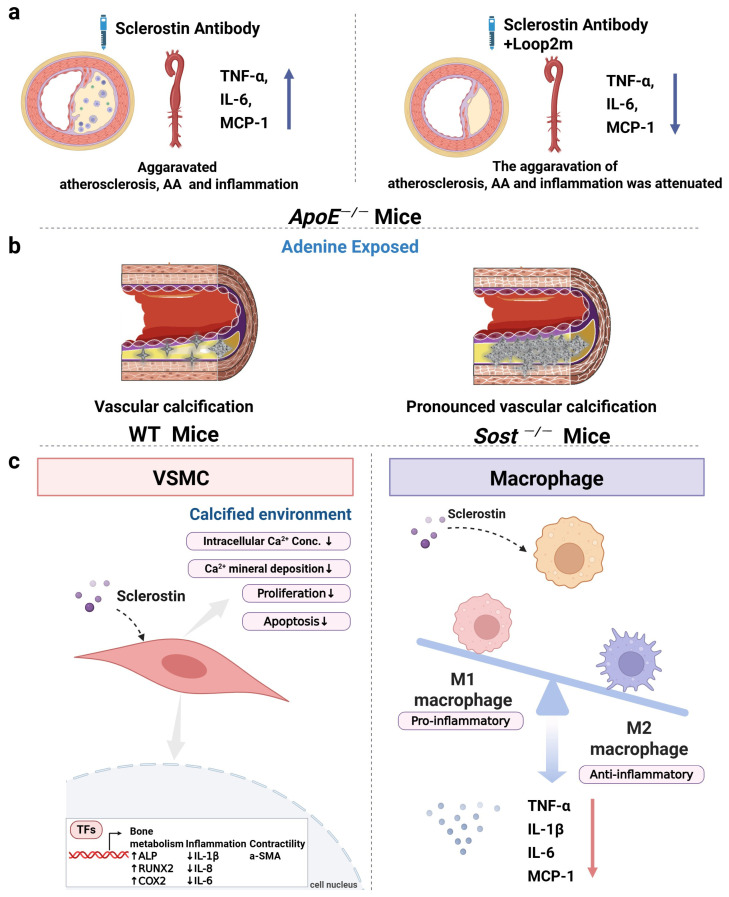
Sclerostin exhibited protective effect on the cardiovascular system. (**a**) The sclerostin antibody induced inhibition of sclerostin’s suppressive effect on atherosclerosis progression (**left**) in *ApoE*^−/−^ mice, while this inhibition was abolished by supplement of exogenous loop2m (**right**). (**b**) Adenine-exposed *Sost*^−/−^ mice exhibited greater calcification in the cardiac vessels (**right**) than adenine-exposed WT mice (**left**). (**c**) Sclerostin in macrophages/VSMCs participated in inhibiting inflammatory responses and vascular calcification development. Sclerostin overexpression in VSMCs decreased calcium deposits in a calcified environment and downregulated the expression of inflammatory genes such as IL1β, IL6, and IL8 (**left**). The addition of sclerostin in macrophages exhibited suppressive effects on inflammatory responses and mediated the conversion of the macrophages into anti-inflammatory M2 phenotypes (**right**). Abbreviations: Loop2m: loop2 mutant; AA: aortic aneurysm. Note: The upward arrows “↑” in the figure represented increase or upregulation. The downward arrows “↓” in the figure represented decrease or downregulation.

**Table 1 ijms-26-04552-t001:** Summary of adverse events associated with the use of romosozumab in FRAME, ARCH, and BRIDGE trials.

Adverse Event	12-Month FRAME Trial [2]		12-Month ARCH Trial [3]		12-Month BRIDGE Trial [4]	
	Placebo (N = 3576)	Romosozumab (N = 3581)	Alendronate (N = 2014)	Romosozumab (N = 2040)	Placebo (N = 81)	Romosozumab (N = 163)
	Number of Patients (Percent)					
Serious adverse event	312(8.7)	344(9.6)	278(13.8)	262(12.8)	10(12.3)	21(12.9)
Events leading to discontinuation of trial regimen	94(2.6)	103(2.9)	64(3.2)	70(3.4)	1(1.2)	5(3.1)
Adjudicated serious cardiovascular event ^a^	15(0.4)	17(0.5)	38(1.9)	50(2.5)	8(4.9)	2(2.5)
Cardiac ischemic event	NA	NA	6(0.3)	16(0.8)	0	3(1.8)
Cerebrovascular event	NA	NA	7(0.3)	16(0.8)	1(1.2)	3(1.8)
Arthralgia	429(12.0)	467(13.0)	NA	NA	NA	NA
Nasopharyngitis	438(12.2)	459(12.8)	218(10.8)	213(10.4)	NA	NA
Back pain	378(10.6)	375(10.5)	228(11.3)	186(9.1)	NA	NA
Injection-site reaction ^b^	104(2.9)	187(5.2)	53(2.6)	90(4.4)	3(3.7)	9(5.5)
Osteonecrosis of the jaw	0	1(<0.1)	0	0	0	0
Atypical femoral fracture	0	1(<0.1)	0	0	0	0

Abbreviations: N, number of subjects who received one or more doses of the drug or placebo; NA, not applicable; ^a^ only included events adjudicated as positive by the independent adjudication committee; ^b^ the most frequent adverse events of injection-site reactions in the romosozumab group during the double-blind period included injection-site pain, erythema, pruritus, hemorrhage, rash, and swelling reactions, which were reported as mild in severity.

**Table 2 ijms-26-04552-t002:** Adverse events associated with the use of romosozumab in post-marketing evaluation.

Adverse Events	Total [6]	Japan	United States
	Romosozumab	Romosozumab	Romosozumab
	(N = 1995)	(N = 1188)	(N = 787)
	Number of Patients (Percent)		
Major cardiovascular event Myocardial infarction	206(10.3)	164(13.8)	41(5.2)
	42(2.1)	28(2.4)	13(1.7)
Stroke Cardiovascular death Other cardiovascular event General cardiac events	84(4.2)	57(4.8)	27(3.4)
	86(4.3)	83(7.0)	<5
	58(2.9)	42(3.5)	16(2.0)
	16(0.8)	10(0.8)	6(0.8)
Bleeding Thrombosis	19(1.0)	19(1.6)	-
	23(1.2)	13(1.1)	10(1.3)

Notes: The combined number of reports from the United States and Japan did not cover the total number of reports. Reports from other countries and regions (N = 20) were not included because the number of events in each cell was too low to calculate.

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
