# Peer review of "Role of Sclerostin in Cardiovascular System"

_ijms, 2025, doi:10.3390/ijms26104552_

Round 1

Reviewer 1 Report

Comments and Suggestions for Authors

I would like to congratulate the authors on this well-structured and comprehensive review. The manuscript addresses a highly relevant and timely topic, presenting a detailed and balanced analysis of both clinical and preclinical evidence regarding the role of sclerostin in the cardiovascular system. The integration of genetic, pharmacological, and mechanistic perspectives adds significant value to the current understanding of this emerging field. The discussion is thoughtful, and the conclusions are well supported by the reviewed literature.

I have only one suggestion to improve the clarity and transparency of the review process:
Please include a brief Methods section or at least a paragraph outlining the literature search strategy, criteria for study inclusion/exclusion, and the databases used. This addition would enhance the methodological rigor of the review and help readers better understand how the evidence was selected and synthesized.

Overall, I find the manuscript scientifically sound and recommend it for publication after this minor revision.

Author Response

Thanks for your valuable comments and suggestions. Please see the details in the attachment.

Reviewer 2 Report

Comments and Suggestions for Authors

Thank you very much for inviting me to review this publication.
The article addresses an important and complex issue by presenting current data, particularly highlighting the controversies and inconsistencies in both clinical and preclinical studies of sclerostin. While sclerostin is well known for its role in bone metabolism (particularly in the context of osteoporosis), its cardiovascular implications are still under investigation.   A publication that combines knowledge of sclerostin's molecular mechanisms (e.g., action in macrophages, endothelium, or smooth muscle cells) with data from clinical trials may help create a more coherent theory about sclerostin's role in the heart and vasculature. It is very favorable that the authors did not limit themselves only to clinical trials, but reached for real-world evidence (real-world data).
However, the review is unclear as to the selection of publications discussed. Please provide a detailed description of how the authors made their selection of publications. The most commonly used in such cases is the PRISMA scheme. 
For example, to my knowledge, an analysis of the STRUCTURE study (comparing romosozumab and teriparatide in postmenopausal women with prior bisphosphonate treatment) would be missing. The main objective of this study was to evaluate the efficacy of romosozumab in the treatment of osteoporosis in postmenopausal women but it evaluated various aspects of safety, including cardiovascular risk.

Comments on the Quality of English Language

The quality of English is satisfactory. It does not limit the understanding of the text.

Author Response

Thanks for your valuable comments and suggestions. Please see the response and revision details in the attachment. Thank you!

Round 2

Reviewer 2 Report

Comments and Suggestions for Authors

Thank you very much for inviting me to review this publication. The authors answered all my questions. They accurately described the methodology, which I cared most about. After the corrections made, I believe that the article is ready for publication. I recommend its acceptance.